# Neighborhood Sampling Does Not Learn the Same Graph Neural Network

## Abstract

Neighborhood sampling is an important ingredient in the training of large-scale graph neural networks. It suppresses the exponential growth of the neighborhood size across network layers and maintains feasible memory consumption and time costs. While it becomes a standard implementation in practice, its systemic behaviors are less understood. We conduct a theoretical analysis by using the tool of neural tangent kernels, which characterize the (analogous) training dynamics of neural networks based on their infinitely wide counterparts—Gaussian processes (GPs). We study several established neighborhood sampling approaches and the corresponding posterior GP. With limited samples, the posteriors are all different, although they converge to the same one as the sample size increases. Moreover, the posterior covariance, which lower-bounds the mean squared prediction error, is uncomparable, aligning with observations that no sampling approach dominates.

## 1 Introduction

Graph neural networks (GNNs) are widely used models (Zhou et al., 2020; Wu et al., 2021) for graph-structured data, such as financial transaction networks, power grids, and molecules and crystals. They encode the relational information present in the data through message passing (Gilmer et al., 2017) on the graph and support a wide array of tasks, including predicting node and graph properties, generating novel graphs, and forecasting interrelated time series. The training of GNNs for large-scale graphs poses a unique challenge in that the computation of the loss of a mini-batch of nodes requires not only their information, but also that of their $L$-hop neighbors due to message passing (also called neighborhood aggregation). The exponential increase of the neighborhood size, especially for power-law graphs, incurs prohibitive memory and time costs and inspires *neighborhood sampling*, a practical mitigation that reduces the neighborhood size and maintains a feasible training cost (Hamilton et al., 2017; Ying et al., 2018; Chen et al., 2018; Zou et al., 2019; Chiang et al., 2019; Zeng et al., 2020). While neighborhood sampling becomes a standard nowadays, its impact on the training behavior remains less understood (Chen & Luss, 2018).

In this work, we conduct a theoretical study on neighborhood sampling by leveraging the emergent tool of neural tangent kernels (NTKs). An NTK (Jacot et al., 2018; Lee et al., 2019) is the dot product of the parameter tangents of a neural network evaluated at two inputs. It was derived from a continuous-time analog of the gradient descent process for network training. This analog—an ordinary differential equation (ODE)—governs the evolution of the network over time given an initial condition. As is well known, an infinitely wide random network is a Gaussian process (GP) (Neal, 1994; Williams, 1996; Lee et al., 2018; de G. Matthews et al., 2018). Hence, using this GP as the initial condition, we obtain its (closed-form) evolution, which is analogous to the training dynamics of the corresponding neural network. For graphs, the NTK becomes a GNTK (Du et al., 2019; Huang et al., 2022; Krishnagopal & Ruiz, 2023) and it governs the evolution of a GNNGP (Niu et al., 2023), which is the infinite-width counterpart of a GNN.

We highlight a few contributions/findings of this work.

1. (Section 3) We first derive the posterior inference for GNNGP under evolution. The posterior GNNGP differs from the (prior) GNNGP, even though both evolve to a limit whose mean interpolates the training nodes. The prior was sporadically studied (Lee et al., 2019), and to our knowledge, its extension to graphs and its posterior are not discussed in the existing literature.

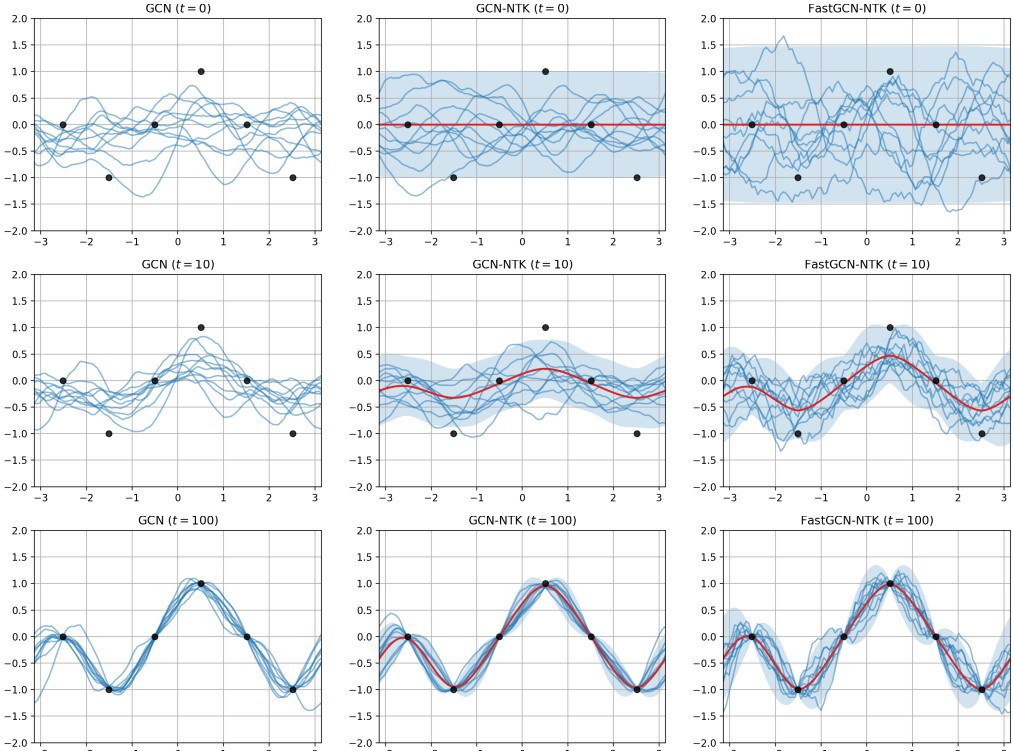

Figure 1: The training dynamics of GCN (left), the evolution of GCN-GP (middle), and that of GCN-GP under layer-wise sampling (right). The black dots are training nodes, the red curves are the GCN-GP means (without or with sampling), and the blue curves are either instances of the GCN being trained or sample paths of the GCN-GP. The shaded region denotes 2x standard deviation. Note that neighborhood sampling drives the GCN-GP to evolve faster to the limit; also note that the sample paths are less smooth. Details of the illustration, including the graph, the training nodes, and the sampling distribution, are given in Section A.

2. (Section 4) Using GCN (Kipf & Welling, 2017) as a working example of GNNs (in which case GNNGP is written as GCN-GP and GNTK is written as GCN-NTK), we analyze two of the most popular neighborhood sampling techniques: layer-wise sampling (Chen et al., 2018), including with and without replacement, and node-wise sampling (Hamilton et al., 2017). We show that in the sampling limit, the prior and posterior GCN-GPs converge to their counterparts without sampling. However, under limited samples, the GCN-GPs resulting from different sampling methods are all different. They appear to be uncomparable and we explain the reasons given some facts of the corresponding covariance matrices and GCN-NTKs. As a consequence, the converged GCN-GPs at the time limit are different, agreeing with the varied performance of trained GNNs observed in practice, when different sampling methods are employed.

3. (Section 5) For a general GNN, we present a programmable approach to composing the GNTK based on its building blocks. This approach extends the composability of NTKs (Yang, 2019; Novak et al., 2020) to the graph case, and more importantly, to neighborhood sampling. We demonstrate its use to derive the GNTK for GraphSAGE (Hamilton et al., 2017), both without and with node-wise sampling (which was the sampling technique proposed in the same paper). Naturally, the findings previewed above for GCN also apply to GraphSAGE.

Figure 1 illustrates the training dynamics of randomly initialized GCNs, the evolution of their infinitely wide counterpart (GCN-GP), and the evolution of GCN-GP under layer-wise sampling (which is the infinitely wide counterpart of FastGCN (Chen et al., 2018)).

## 2 BACKGROUND: INFINITELY WIDE GNN AND THE GNTK

As a necessary background for the analysis of neighborhood sampling in GNN training, this section extends NTK to GNTK. While we are not the first to study GNTKs, existing work (Du et al., 2019; Huang et al., 2022; Krishnagopal & Ruiz, 2023) focused on a certain GNN architecture that is akin to GIN (Xu et al., 2019). In contrast, we embrace more architectures, particularly those tied to the proposals of neighborhood sampling. Specifically, we use the GCN as a working example in this section and extend the analysis for other GNNs (e.g., GraphSAGE) in Section 5, by introducing the "kernel transformation" technique that treats a GNN as a composition of basic building blocks.

### 2.1 INFINITELY WIDE GCN

Denote by $\mathcal{G} = (\mathcal{V}, \mathcal{E})$ a graph with $N = |\mathcal{V}|$ nodes and $M = |\mathcal{E}|$ edges. For notational simplicity, we use $A \in \mathbb{R}^{N \times N}$ to mean a normalized graph adjacency matrix, allowing any form of normalization. Using $d_l$ to denote the width of the $l$-th layer, the layer architecture of GCN reads

$$\text{GCN:} \qquad X^{(l)} = \phi(Z^{(l)}) \equiv \phi\left(\frac{\sigma_w}{\sqrt{d_{l-1}}} A X^{(l-1)} W^{(l)} + \sigma_b \mathbf{1}_{N \times 1} b^{(l)}\right), \qquad (1)$$

where $X^{(l-1)} \in \mathbb{R}^{N \times d_{l-1}}$ and $X^{(l)} \in \mathbb{R}^{N \times d_l}$ are layer inputs and outputs, respectively; $W^{(l)} \in \mathbb{R}^{d_{l-1} \times d_l}$ and $b^{(l)} \in \mathbb{R}^{1 \times d_l}$ are the weights and biases; and $\phi$ is the activation function. Eqn. (1) differs slightly from the standard GCN by explicitly using the factors $\sigma_w / \sqrt{d_{l-1}}$ and $\sigma_b$ to scale the weights and biases, so that the subsequent formulas for GNTK is neater. We will frequently operate on a single feature dimension and a single node of the layer. Hence, we write

$$x_i^{(l)}(x) = \phi(z_i^{(l)}(x)), \quad z_i^{(l)}(x) = \sum_{v \in \mathcal{V}} A_{xv} y_i^{(l)}(v) + \sigma_b b_i^{(l)}, \quad y_i^{(l)}(x) = \frac{\sigma_w}{\sqrt{d_{l-1}}} \sum_{j=1}^{d_{l-1}} W_{ji}^{(l)} x_j^{(l-1)}(x),$$

$$(2)$$

where, for example, $x_i^{(l)}(x)$ denotes the $i$-th feature for the node $x$ in the post-activation $X^{(l)}$, while $z_i^{(l)}(x)$ is the corresponding pre-activation.

The following theorem establishes the GCN-GP and its recursive computation. It is equivalent to Niu et al. (2023, Theorem 1).

**Theorem 1.** *For an $L$-layer GCN, assume $d_1, \ldots, d_{L-1}$ to be infinite in succession and let the bias $b_i^{(l)}$ and the weight $W_{ij}^{(l)}$ be independent standard normal, for all $i$, $j$, and $l$. Then, for each $i$, the collection $\{z_i^{(l)}(x)\}$ over all graph nodes $x$ follows the normal distribution $\mathcal{N}(0, K^{(l)})$, where the covariance matrix $K^{(l)}$ can be computed recursively by*

$$K^{(l)} = \sigma_b^2 \mathbf{1}_{N \times N} + \sigma_w^2 A C^{(l-1)} A^T, \qquad (3)$$

$$C^{(l)} = \mathrm{E}_{z_i^{(l)} \sim \mathcal{N}(0, K^{(l)})}[\phi(z_i^{(l)}) \phi(z_i^{(l)})^T], \qquad (4)$$

*with $C^{(0)} = X^{(0)} (X^{(0)})^T / d_0$.*

Throughout, we assume that the GCN performs a scalar-output regression for each node. In this case, $d_L = 1$ and the final layer does not have an activation. In other words, the GCN output is $Z^{(L)}$ (rather than $X^{(L)}$) and it has a single column. The covariance matrix of the GCN-GP is $K = K^{(L)}$.

### 2.2 GRAPH NEURAL TANGENT KERNEL

A typical gradient descent algorithm reads $\theta' = \theta - \eta \nabla_\theta \mathcal{L}$, where $\mathcal{L}$, $\theta$, and $\eta$ are the training loss, the training parameters, and the learning rate, respectively. A continuous-time analog of gradient descent is thus $\frac{d}{dt}\theta(t) = -\eta \nabla_\theta \mathcal{L} = -\eta (\nabla_\theta f)^T (\nabla_f \mathcal{L})$, where recall that the loss is a function of the network $f(\theta) : \mathbb{R}^P \to \mathbb{R}^N$ for $P$ parameters and $N$ samples (we denote $\nabla_\theta f \in \mathbb{R}^{N \times P}$). This formula leads to the following ODE

$$\frac{df(t)}{dt} = (\nabla_\theta f)\frac{d\theta(t)}{dt} = -\eta[(\nabla_\theta f)(\nabla_\theta f)^T](\nabla_f \mathcal{L}), \qquad (5)$$

by a simple invocation of the chain rule. This equation describes the evolution of the network $f(\theta(t))$ over time given an initial condition at $t = 0$. The NKT $\Theta(\theta)$ is defined as

$$\Theta(\theta(t)) := (\nabla_\theta f)(\nabla_\theta f)^T \in \mathbb{R}^{N \times N}. \tag{6}$$

In words, the kernel is the inner product of the gradients of $f$ evaluated at a pair of samples.

While the NTK has a time dependence, for infinitely many random parameters, the inner product is a constant independent of $t$. For graphs, we derive in the following theorem that computes the constant GCN-NTK $\Theta = \Theta^{(L)}$ through layer-by-layer recursion.

**Theorem 2.** *Under the condition and notations of Theorem 1, the neural tangent kernel $\Theta^{(l)}$ can be computed recursively by*

$$\Theta^{(l)} = \sigma_b^2 \mathbf{1}_{N \times N} + \sigma_w^2 A V^{(l-1)} A^T, \tag{7}$$

$$V^{(l-1)} = \Theta^{(l-1)} \odot \dot{C}^{(l-1)} + C^{(l-1)}, \tag{8}$$

$$\dot{C}^{(l)} = \mathrm{E}_{z_i^{(l)} \sim \mathcal{N}(0, K^{(l)})}[\dot{\phi}(z_i^{(l)})\dot{\phi}(z_i^{(l)})^T], \tag{9}$$

*with $\dot{C}^{(0)} = \mathbf{0}_{N \times N}$ and $\Theta^{(0)} = \mathbf{0}_{N \times N}$, where $\dot{\phi}$ denotes the derivative of the activation function $\phi$.*

All proofs of this paper are given in Section B. Theorems 1 and 2 involve the second raw moment of the post-activation $\phi(z_i^{(l)})$ and that of its derivative $\dot{\phi}(z_i^{(l)})$. The formulas are given in Section C for some activation functions.

## 3 EVOLUTION OF INFINITELY WIDE GNN

The constant kernel $\Theta$ leads to a closed-form solution $f(t)$, which is analogous to the training dynamics of a GNN. When the initial condition $f(0)$ is a GP that follows $\mathcal{N}(0, K(0))$ for $K(0) \equiv K^{(L)}$ defined in Theorem 1, we can obtain the GP at time $t$ as $\mathcal{N}(\mu(t), K(t))$. The discussions here apply to any GNN, although they reference $K^{(L)}$ and $\Theta^{(L)}$ from Theorems 1 and 2 for GCNs.

Specifically, let subscripts $b$ and $c$ denote the training set and the remaining set (e.g., the prediction set), respectively. Then, the GCN output $f$ is split in two parts, $f_b$ and $f_c$. When we use the squared loss $\mathcal{L} = \frac{1}{2N_b}\|f_b(t) - y_b\|_2^2$ for regression, the ODE becomes

$$\frac{d}{dt}\begin{pmatrix} f_b(t) \\ f_c(t) \end{pmatrix} = -\eta\Theta\nabla_f\mathcal{L} = -\frac{\eta}{N_b}\begin{pmatrix} \Theta_{bb} & \Theta_{bc} \\ \Theta_{cb} & \Theta_{cc} \end{pmatrix}\begin{pmatrix} f_b(t) - y_b \\ 0 \end{pmatrix}. \tag{10}$$

Following this ODE, we obtain the joint distribution of $f_b(t)$ and $f_c(t)$.

**Theorem 3.** *Under the conditions and notations of Theorems 1 and 2, the infinitely wide GCN $f$ is a GP that evolves over time $t \geq 0$ as*

$$\begin{pmatrix} f_b(t) \\ f_c(t) \end{pmatrix} \sim \mathcal{N}\left(\begin{pmatrix} \mu_b(t) \\ \mu_c(t) \end{pmatrix}, \begin{pmatrix} K_{bb}(t) & K_{bc}(t) \\ K_{cb}(t) & K_{cc}(t) \end{pmatrix}\right),$$

*where*

$$\mu_b(t) = \beta y_b, \qquad \mu_c(t) = \Theta_{cb}\Theta_{bb}^{-1}\beta y_b,$$

$$K_{bb}(t) = \alpha K_{bb}(0)\alpha, \qquad K_{cb}(t) = K_{cb}(0)\alpha - \Theta_{cb}\Theta_{bb}^{-1}\beta K_{bb}(0)\alpha = K_{bc}(t)^T, \tag{11}$$

$$K_{cc}(t) = K_{cc}(0) - \Theta_{cb}\Theta_{bb}^{-1}\beta K_{bc}(0) - K_{cb}(0)\beta\Theta_{bb}^{-1}\Theta_{bc} + \Theta_{cb}\Theta_{bb}^{-1}\beta K_{bb}(0)\beta\Theta_{bb}^{-1}\Theta_{bc},$$

*with $\alpha = \exp(-t\eta\Theta_{bb}/N_b)$ and $\beta = I - \alpha$.*

The above result indicates that the converged GCN-GP has mean and covariance, respectively,

$$\mu_c(\infty) = \Theta_{cb}\Theta_{bb}^{-1}y_b,$$

$$K_{cc}(\infty) = K_{cc}(0) - \Theta_{cb}\Theta_{bb}^{-1}K_{bc}(0) - K_{cb}(0)\Theta_{bb}^{-1}\Theta_{bc} + \Theta_{cb}\Theta_{bb}^{-1}K_{bb}(0)\Theta_{bb}^{-1}\Theta_{bc}. \tag{12}$$

Note that the mean $\mu_c(\infty)$ reads like the posterior mean of a GP with $\Theta$ being the covariance matrix, but the covariance $K_{cc}(\infty)$ differs from the posterior covariance of such a GP (which is $\Theta_{cc} - \Theta_{cb}\Theta_{bb}^{-1}\Theta_{bc}$). Hence, the NTK $\Theta$ does not admit a covariance kernel interpretation.

**Posterior inference.** What Theorem 3 offers is a prior characterization of the GCN-GP $f(t)$. One may use this prior to perform posterior inference by assuming observation $y_b$ at all times:

$$\mathrm{E}[f_c(t)|f_b(t) \equiv y_b] = \mu_c(t) + K_{cb}(t)K_{bb}(t)^{-1}(y_b - \mu_b(t)).$$

However, this formula is evaluated to a constant $K_{cb}(0)K_{bb}(0)^{-1}y_b$, which means that the posterior GP (at least the mean, but could also be shown for the covariance) never evolves. The theoretical pitfall originates from performing GP posterior inference without assuming noise for the observation. To fix the pitfall, we assume that the observations have iid Gaussian noise with variance $\epsilon > 0$. A benefit of this assumption is that it can rescue the degenerate case when the covariance matrix $K_{bb}(0)$ is rank-deficient. In this case, the corrected posterior mean is

$$\begin{aligned}
\mathrm{E}[f_c(t)|f_b(t) \equiv y_b] &= \mu_c(t) + K_{cb}(t)[K_{bb}(t) + \epsilon I]^{-1}(y_b - \mu_b(t)) \\
&= \Theta_{cb}\Theta_{bb}^{-1}\beta y_b + K_{cb}(t)[K_{bb}(t) + \epsilon I]^{-1}\alpha y_b,
\end{aligned} \tag{13}$$

and the posterior covariance becomes

$$\mathrm{Cov}[f_c(t)|f_b(t) \equiv y_b] = K_{cc}(t) - K_{cb}(t)[K_{bb}(t) + \epsilon I]^{-1}K_{bc}(t). \tag{14}$$

Interestingly, at the time limit $t = \infty$, the posterior mean coincides with the prior mean $\mu_c(\infty) = \Theta_{cb}\Theta_{bb}^{-1}y_b$, which interpolates the training data $y_b$, and the posterior covariance coincides with the prior covariance $K_{cc}(\infty)$, too. Additionally, at any time $t$, the posterior covariance is always no greater than the prior one in the Loewner order.

## 4 GCN-NTK Under Neighborhood Sampling

Neighborhood sampling is an important training ingredient unique to GNNs. Different from the training of usual networks, where a mini-batch of size $B$ involves only $B$ data points, a GNN training step involves in the worst case $O(Bd^L)$ graph nodes for a degree-$d$ graph and an $L$-layer GNN, because in every layer, the node feature is updated by aggregating the features of its $O(d)$ neighbors. Such an exponential growth of the neighborhood size renders prohibitive memory and time costs for even shallow GNNs. Neighborhood sampling is a collection of techniques that reduce the neighborhood size and maintain feasible training costs. In this section, we analyze the impact of neighborhood sampling on the (analogous) training dynamics of GCN. In particular, we consider layer-wise sampling and node-wise sampling, which are amenable to an analysis of $K^{(L)}$ and $\Theta^{(L)}$.

### 4.1 Layer-Wise Sampling With Replacement

FastGCN (Chen et al., 2018) proposes layer-wise sampling, which admits variants such as LADIES (Zou et al., 2019). In such a technique, a set of nodes, $\mathcal{V}_l \subset \mathcal{V}$, is sampled and only nodes in $\mathcal{V}_l$ will participate in the neighborhood aggregation in the $l$-th layer. Because fewer nodes are aggregated, their contributions requires rescaling. The total number of involved nodes is at most $B + \sum_{l=1}^{L} |\mathcal{V}_l|$, suppressing the exponential growth of the neighborhood size without sampling. Formally, the procedure is defined in the following.

**Definition 1** (Layer-wise sampling with replacement). Given a probability distribution $p_l$ over the node set $\mathcal{V}$ (that is, $\sum_{v \in \mathcal{V}} p_l(v) = 1$) for each layer $l = 1, \ldots, L$, sample $N_l$ nodes with replacement and scale each sample $v$ with $1/p_l(v)$ when performing the neighborhood aggregation $AY^{(l)}$.

Definition 1 is essentially a form of importance sampling by using the proposal distribution $p_l$. FastGCN suggests setting $p_l(v)$ to be proportional to the squared 2-norm of $A(:, v)$. Sampling with replacement allows one to express the neighborhood aggregation $AY^{(l)}$ as an expectation involving $N_l$ categorical variables. Specifically, if we use $w^j$ to denote a random one-hot vector following $w^j \sim \mathrm{Cat}(p_l)$ for $j = 1, \ldots, N_l$ and let each $D_l^j$ be the probability-scaled diagonal matrix $D_l^j = \mathrm{diag}(w^j/p_l)$, then

$$AY^{(l)} = \mathrm{E}_{w^1, \ldots, w^{N_l}} \left[ \frac{1}{N_l} \sum_{j=1}^{N_l} A D_l^j Y^{(l)} \right]. \tag{15}$$

This linear expectation is key to the analysis of $K^{(L)}$ and $\Theta^{(L)}$, while the computation formulas for $C^{(l-1)}$ and $V^{(l-1)}$ resulting from the handling of nonlinear activation functions remain unchanged, as the following result states.

**Theorem 4.** *Under layer-wise sampling with replacement, the covariance matrix and the neural tangent kernel become, respectively,*

$$K^{(l)} = \sigma_b^2 \mathbf{1}_{N \times N} + \sigma_w^2 A(M_p^{(l)} \odot C^{(l-1)}) A^T, \tag{16}$$

$$\Theta^{(l)} = \sigma_b^2 \mathbf{1}_{N \times N} + \sigma_w^2 A(M_p^{(l)} \odot V^{(l-1)}) A^T, \tag{17}$$

*where the matrix $M_p^{(l)} \in \mathbb{R}^{N \times N}$ has elements*

$$M_p^{(l)}(v, v') = \begin{cases} 1 + \frac{1 - p_l(v)}{p_l(v) N_l}, & v = v' \\ 1 - \frac{1}{N_l}, & v \neq v' \end{cases} \tag{18}$$

*and $C^{(l-1)}$ and $V^{(l-1)}$ follow Eqn. (4), (8), and (9), recursively computed by using the $K^{(l-1)}$ and $\Theta^{(l-1)}$ in this theorem.*

Theorem 4 suggests that the initial GCN-GP has a covariance matrix $K^{(L)}$, which, when computed recursively layer-by-layer, is modified by using a *masking matrix $M_p^{(l)}$* applied to the second raw moment matrix $C^{(l-1)}$ of (a column of) the layer input $X^{(l-1)} = \phi(Z^{(l-1)})$. This modification leads to a change of the posterior inference, including the mean $K_{cb}^{(L)}(K_{bb}^{(L)} + \epsilon I)^{-1} y_b$ and the covariance $K_{cc}^{(L)} - K_{cb}^{(L)}(K_{bb}^{(L)} + \epsilon I)^{-1} K_{bc}^{(L)}$. Nevertheless, as the number of samples, $N_l$, increases, $M_p^{(l)}$ tends to the matrix of all ones for a fixed sampling distribution $p_l$. Then, by continuity, $K^{(L)}$ converges to the covariance matrix without sampling. In other words, in the sampling limit, the initial GCN-GP under layer-wise sampling with replacement converges to the initial GCN-GP without sampling.

The initial GCN-GP evolves according to the training dynamics laid out by the ODE (10). At any time $t$, the GCN-GP has a prior mean $\mu_c(t)$ and a prior covariance $K_{cc}(t)$ following (11), as well as a posterior mean $\mathbb{E}[f_c(t)|f_b(t) \equiv y_b]$ and a posterior covariance $\text{Cov}[f_c(t)|f_b(t) \equiv y_b]$ following (13) and (14), respectively. All these quantities involve not only $K^{(L)}$ but also the GCN-NTK $\Theta^{(L)}$. Similar to $K^{(L)}$, the modification of $\Theta^{(L)}$ is recursively layer-by-layer, by using the same masking matrix $M_p^{(l)}$ for each layer. As sample size increases, all the modified quantities tend to the counterpart without sampling.

To put the value of Theorem 4 in context, we compare it with the analysis of FastGCN by Chen & Luss (2018). The referenced work studies the training of FastGCN from the angle of the gradient. It points out that the stochastic gradient under layer-wise sampling is biased (which is not surprising because expectation cannot be exchanged with nonlinear activation functions), but it is consistent because the stochastic gradient converges to the true gradient in probability as $N_l \to \infty$ for all $l$. Consistent gradient can drive gradient-descent training to convergence in the sense that the stochastic gradient can have an arbitrarily small norm. It is, however, unclear what the converged GCN is and how it is connected with the one without layer-wise sampling. In contrast, our analysis points out that for infinitely wide GCN (which becomes a GP), the converged GCN-GP has a posterior mean $\Theta_{cb}\Theta_{bb}^{-1} y_b$ and posterior covariance $K_{cc}(\infty)$. Sampling and no sampling are connected in that the posterior for the former converges to that for the latter in the sampling limit.

### 4.2 LAYER-WISE SAMPLING WITHOUT REPLACEMENT

A consequence of sampling with replacement is that neighborhood aggregation will aggregate neighbors with multiplicity. An alternative is to perform sampling without replacement, which admits an implementation convenience. However, when the nodes have nonuniform sampling probabilities, the analysis is challenging. Hence, we analyze a variant where each node is sampled independently. This variant also results in non-repetitive samples, but one cannot pre-specify the desired number of samples, $N_l$. Instead, one controls its expected number through the sampling probabilities.

**Definition 2** (Layer-wise sampling without replacement). For each layer $l = 1, \ldots, L$, given $q_l(v) \in (0, 1]$ for each node $v$, sample $v$ with probability $q_l(v)$ independently and scale each sample with $1/q_l(v)$ when performing neighborhood aggregation.

In this case, $K^{(L)}$ and $\Theta^{(L)}$ follow a similar modification to that in the preceding subsection, but using a different masking matrix.

**Theorem 5.** *Under layer-wise sampling without replacement, the covariance matrix and the neural tangent kernel become, respectively,*

$$K^{(l)} = \sigma_b^2 \mathbf{1}_{N \times N} + \sigma_w^2 A(M_q^{(l)} \odot C^{(l-1)})A^T, \tag{19}$$

$$\Theta^{(l)} = \sigma_b^2 \mathbf{1}_{N \times N} + \sigma_w^2 A(M_q^{(l)} \odot V^{(l-1)})A^T, \tag{20}$$

*where the matrix $M_q^{(l)} \in \mathbb{R}^{N \times N}$ has elements*

$$M_q^{(l)}(v, v') = \begin{cases} 1/q_l(v), & v = v' \\ 1, & v \neq v' \end{cases} \tag{21}$$

*and $C^{(l-1)}$ and $V^{(l-1)}$ follow Eqn. (4), (8), and (9), recursively computed by using the $K^{(l-1)}$ and $\Theta^{(l-1)}$ in this theorem.*

### 4.3 NODE-WISE SAMPLING

Another popular neighborhood sampling approach is node-wise sampling, advocated by the Graph-SAGE authors (Hamilton et al., 2017). Here, we apply it to GCN so that discussions are more coherent. Its application to GraphSAGE will be discussed in Section 5. In this approach, only sampling without replacement makes sense. With a budget $k$ in mind, at most $k$ neighbors are sampled from a node $x$. Similar to the reasoning in the preceding subsection, we analyze a variant where each neighbor is independently sampled. Moreover, the sampling probabilities are uniform, which agree with practice. The procedure is formally defined in the following.

**Definition 3** (Node-wise sampling)**.** Let $\mathcal{N}(x)$ denote $x$'s neighborhood, which may include $x$ itself if $A(x, x) \neq 0$. Given a positive integer $k$ (called the "fanout"), define a probability distribution $q_x$, where

$$q_x(v) := \begin{cases} \frac{k}{|\mathcal{N}(x)|}, & |\mathcal{N}(x)| \geq k \text{ and } v \in \mathcal{N}(x) \\ 1, & |\mathcal{N}(x)| < k \text{ and } v \in \mathcal{N}(x) \\ 0, & v \notin \mathcal{N}(x). \end{cases}$$

For each node $x$, sample $v$ with probability $q_x(v)$ independently and scale each sample with $1/q_x(v)$ when performing neighborhood aggregation. Here, we have dropped the layer index $l$ in $q_x$ and $k$ to avoid notation cluttering.

**Theorem 6.** *Under node-wise sampling, the covariance matrix and the neural tangent kernel become, respectively,*

$$K^{(l)}(x, x') = \sigma_b^2 + \sigma_w^2 A(x, :)(M_{xx'}^{(l)} \odot C^{(l-1)})A(x', :)^T, \tag{22}$$

$$\Theta^{(l)}(x, x') = \sigma_b^2 + \sigma_w^2 A(x, :)(M_{xx'}^{(l)} \odot V^{(l-1)})A(x', :)^T, \tag{23}$$

*where the matrix $M_{xx'}^{(l)} \in \mathbb{R}^{N \times N}$ has elements*

$$\text{when } x = x': \quad M_{xx}^{(l)}(v, v') = \begin{cases} 1/q_x(v), & v, v' \in \mathcal{N}(x) \text{ and } v = v' \\ 1, & v, v' \in \mathcal{N}(x) \text{ and } v \neq v' \\ 0, & \text{otherwise} \end{cases} \tag{24}$$

$$\text{when } x \neq x': \quad M_{xx'}^{(l)}(v, v') = \begin{cases} 1, & v \in \mathcal{N}(x) \text{ and } v' \in \mathcal{N}(x') \\ 0, & \text{otherwise} \end{cases} \tag{25}$$

*and $C^{(l-1)}$ and $V^{(l-1)}$ follow Eqn. (4), (8), and (9), recursively computed by using the $K^{(l-1)}$ and $\Theta^{(l-1)}$ in this theorem.*

### 4.4 DISCUSSIONS

All three neighborhood sampling approaches analyzed so far use masking matrices to modify $K^{(l)}$ and $\Theta^{(l)}$ layer by layer. Despite their differences, all masking matrices admit a notion of convergence in the sampling limit. For layer-wise sampling with replacement, $M_p^{(l)} \to \mathbf{1}_{N \times N}$ when the sample size $N_l \to \infty$. For layer-wise sampling without replacement, $M_q^{(l)} \to \mathbf{1}_{N \times N}$ when the sampling

probability $q_l(v) \to 1$ for all nodes $v$. For node-wise sampling, in expectation more and more neighbors are sampled when the fanout $k \to \max_x |\mathcal{N}(x)|$, because in this case $q_x(v) \to 1$ for all $x$ and $v$. In the limit, based on mathematical induction, $K^{(L)}$ and $\Theta^{(L)}$ converge to their counterparts without sampling, and hence by continuity, all relevant quantities converge to their counterparts without sampling. The following theorem summarizes this obvious result.

**Theorem 7.** *For all neighborhood sampling approaches presented in Sections 4.1–4.3, as the sample size increases, the covariance matrix $K^{(L)}$, the neural tangent kernel $\Theta^{(L)}$, the prior GCN-GP $f(t)$ at any time $t$, and the posterior $f_c(t)|f_b(t)$ converge to the their counterparts without sampling.*

While the above result indicates that there is virtually no difference among the sampling approaches with sufficiently large samples, under limited samples, the resulting prior/posterior GCN-GPs are different. We wish to compare them and seek the best sampling approach. For example, it is known that the posterior covariance is a lower bound of the mean squared prediction error for GPs (Wagberg et al., 2017). We wish to find the smallest posterior covariance. Unfortunately, they do not seem to be comparable. To elucidate this, we gather a few facts for the masking matrices $M_p^{(l)}$ and $M_q^{(l)}$.

**Proposition 8.** *The following properties hold. (i) $M_p^{(l)} \succeq \mathbf{1}_{N \times N}$. (ii) $M_q^{(l)} \succeq \mathbf{1}_{N \times N}$. (iii) For any symmetric positive semi-definite matrix $B$, $M_p^{(l)} \odot B \succeq B$ and $M_q^{(l)} \odot B \succeq B$. (iv) When $q_l = N_l p_l$, $M_p^{(l)} - M_q^{(l)}$ has one eigenvalue equal to $1 - \frac{N}{N_l} < 0$ and $N - 1$ eigenvalues equal to $1$.*

Proposition 8 offers some hints as to why the covariance matrices $K^{(L)}$ are uncomparable among sampling methods. The expression $K^{(l)} = \sigma_b^2 \mathbf{1}_{N \times N} + \sigma_w^2 A(M_p^{(l)} \odot C^{(l-1)})A^T$ (see (16)) appears to suggest that $K^{(l)}$ for layer-wise sampling with replacement is greater than, in the Loewner order, its counterpart without sampling based on Property (iii). However, the expression is recurrent and $C^{(l-1)}$ is a function of $K^{(l-1)}$. In general, it does not hold that $C_1^{(l-1)} \succeq C_2^{(l-1)}$ when $K_1^{(l-1)} \succeq K_2^{(l-1)}$, even for the most common ReLU activation (see Section D). Hence, an attempt of mathematical induction fails. As a consequence, we cannot show that $K^{(L)}$ under layer-wise sampling is greater than its counterpart without sampling. Similarly, a Loewner order between $K^{(L)}$ under layer-wise sampling with replacement and that without replacement cannot be established, because there is no such order between $M_p^{(l)}$ and $M_q^{(l)}$ (see Property (iv)). That the three sampling approaches are uncomparable agrees with practical observations when training GNNs.

## 5 FROM GCN TO GENERAL GNNS

As we have seen, the GNTK theory, its posterior inference, and the convergence of neighborhood sampling are largely independent of the GNN architecture, but the specific computation uses $K^{(L)}$ and $\Theta^{(L)}$ that are architecture-dependent. For a new GNN, one may rework their formulas like those in Theorems 1 and 2. The reworking, in fact, can be effortless by noting how these theorems are proved: a layer is composed of building blocks, each of which incurs a corresponding transformation for the covariance and the NTK. Such a nice property allows one to easily derive the recursion formulas for $K^{(l)}$ and $\Theta^{(l)}$ like writing a GNN program and obtaining a transformation of it automatically through operator overloading (Yang, 2019; Novak et al., 2020). We call the covariance matrices and the NTKs *programmable*. Niu et al. (2023) provided the covariance transformations; here, we complement them with the NTK transformations and neighborhood sampling in Table 1.

We use Table 1 to demonstrate the derivation of $K^{(l)}$ and $\Theta^{(l)}$ for GraphSAGE:

$$\text{GraphSAGE:} \quad X^{(l)} = \phi\left(\frac{\sigma_w}{\sqrt{d_{l-1}}} X^{(l-1)} W_1^{(l)} + \frac{\sigma_w}{\sqrt{d_{l-1}}} A X^{(l-1)} W_2^{(l)} + \sigma_b \mathbf{1}_{N \times 1} b^{(l)}\right). \quad (26)$$

To apply the transformations, we use $X$ to denote pre-activation rather than post-activation as in (26) and remove the layer index for simplicity: $X \leftarrow \frac{\sigma_w}{\sqrt{d}} \phi(X) W_1 + \frac{\sigma_w}{\sqrt{d}} A\phi(X) W_2 + \sigma_b \mathbf{1}_{N \times 1} b$. This layer has four parts: (i) activation $\phi(X)$; (ii) linear transformation for each node $\frac{\sigma_w}{\sqrt{d}} \phi(X) W_1$; (iii) graph convolution $\frac{\sigma_w}{\sqrt{d}} A\phi(X) W_2$; and (iv) bias $\sigma_b \mathbf{1}_{N \times 1} b$. Part (i) transforms the covariance to $K \leftarrow g(K)$ and the NTK to $\Theta \leftarrow \Theta \odot h(K)$. Then, part (ii) yields $K \leftarrow \sigma_w^2 g(K)$ and $\Theta \leftarrow$

Table 1: GNN building blocks and covariance and GNTK operations. $g(K) := C$ and $h(K) := \dot{C}$.

| Building block | Neural network | Covariance operation | GNTK operation |
|---|---|---|---|
| Input | $X \leftarrow X^{(0)}$ | $K \leftarrow C^{(0)}$ | $\Theta \leftarrow \mathbf{0}_{N \times N}$ |
| Bias term | $X \leftarrow X + \sigma_b \mathbf{1}_{N \times 1} b$ | $K \leftarrow K + \sigma_b^2 \mathbf{1}_{N \times N}$ | $\Theta \leftarrow \Theta + \sigma_b^2 \mathbf{1}_{N \times N}$ |
| Weight term | $X \leftarrow \frac{\sigma_w}{\sqrt{d}} XW$ | $K \leftarrow \sigma_w^2 K$ | $\Theta \leftarrow \sigma_w^2 \Theta + \sigma_w^2 K$ |
| Mixed weight term | $X \leftarrow X(\alpha I + \frac{\beta \sigma_w}{\sqrt{d}} W)$ | $K \leftarrow (\alpha^2 + \beta^2 \sigma_w^2)K$ | $\Theta \leftarrow (\alpha^2 + \beta^2 \sigma_w^2)\Theta + \beta^2 \sigma_w^2 K$ |
| Graph convolution | $X \leftarrow AX$ | $K \leftarrow AKA^T$ | $\Theta \leftarrow A\Theta A^T$ |
| Activation | $X \leftarrow \phi(X)$ | $K \leftarrow g(K)$ | $\Theta \leftarrow \Theta \odot h(K)$ |
| Independent addition | $X \leftarrow X_1 + X_2$ | $K \leftarrow K_1 + K_2$ | $\Theta \leftarrow \Theta_1 + \Theta_2$ |
| Layer-wise sampling | $X \leftarrow ADX$ | $K \leftarrow A(M \odot K)A^T$ | $\Theta \leftarrow A(M \odot \Theta)A^T$ |
| Node-wise sampling | $X(x,:) \leftarrow A(x,:)D_x X$ | $K(x,x') \leftarrow A(x,:)(M_{xx'} \odot K)A(x',:)^T$ | |
| | | $\Theta(x,x') \leftarrow A(x,:)(M_{xx'} \odot \Theta)A(x',:)^T$ | |

$\sigma_w^2(\Theta \odot h(K) + g(K))$. Next, part (iii) yields a transformation which, on top of part (ii), multiplies $A$ to the left and $A^T$ to the right. Finally, adding (ii) and (iii) because they are independent, followed by adding the bias in part (iv), we obtain the overall transformation summarized below.

**Theorem 9.** *Under the conditions and notations of Eqn. (4), (8), and (9), for GraphSAGE, let the two weight terms $W_1^{(l)}$ and $W_2^{(l)}$ follow the same distribution as $W^{(l)}$ (i.e., standard normal). Then, the covariance matrix and the neural tangent kernel become, respectively,*

$$K^{(l)} = \sigma_b^2 \mathbf{1}_{N \times N} + \sigma_w^2 C^{(l-1)} + \sigma_w^2 A C^{(l-1)} A^T,$$

$$\Theta^{(l)} = \sigma_b^2 \mathbf{1}_{N \times N} + \sigma_w^2 V^{(l-1)} + \sigma_w^2 A V^{(l-1)} A^T,$$

*where $C^{(l-1)}$ and $V^{(l-1)}$ follow Eqn. (4), (8), and (9), recursively computed by using the $K^{(l-1)}$ and $\Theta^{(l-1)}$ in this theorem.*

Table 1 includes the transformations for neighborhood sampling. Layer-wise sampling reads $X \leftarrow ADX$, where $D$ is a diagonal matrix including sample indicator and probability scaling (see (15)). For node-wise sampling, each node $x$ maintains a different sampling distribution and hence the transformation is dependent on the $(x, x')$ pair. By applying the last row of Table 1, we have:

**Theorem 10.** *Under node-wise sampling, for GraphSAGE, the covariance matrix and the neural tangent kernel become, respectively,*

$$K^{(l)}(x, x') = \sigma_b^2 + \sigma_w^2 C^{(l-1)}(x, x') + \sigma_w^2 A(x,:)(M_{xx'}^{(l)} \odot C^{(l-1)})A(x',:)^T,$$

$$\Theta^{(l)}(x, x') = \sigma_b^2 + \sigma_w^2 V^{(l-1)}(x, x') + \sigma_w^2 A(x,:)(M_{xx'}^{(l)} \odot V^{(l-1)})A(x',:)^T,$$

*where $C^{(l-1)}$ and $V^{(l-1)}$ follow Eqn. (4), (8), and (9), recursively computed by using the $K^{(l-1)}$ and $\Theta^{(l-1)}$ in this theorem, and $M_{xx'}^{(l)}$ is defined in Theorem 6.*

## 6 ADDITIONAL RELATED WORK AND CONCLUDING REMARKS

An additional school of related work that has not been discussed includes the extension of the GP and NTK from feed-forward networks to modern architectures such as convolution layers (Novak et al., 2019), recurrent networks (Yang, 2019), and residual connections (Garriga-Alonso et al., 2019). Some popular architectures/building blocks are too complex and their fit to Table 1 requires in-depth investigations. A notable example is the attention layer (Hron et al., 2020), which has a popular graph counterpart: the graph attention network, GAT (Veličković et al., 2018). However, analyzing GAT is faced with many complications: (i) it uses additive attention rather than the more common multiplicative ones nowadays; (ii) the study by Hron et al. (2020) either uses the same query weights and key weights for $d^{-1}$ scaling or removes the softmax for $d^{-\frac{1}{2}}$ scaling; and (iii) it is challenging to fit neighborhood sampling into the analysis framework.

A curious observation (e.g., from Figure 1) is that neighborhood sampling appears to incur a larger prior/posterior covariance than does no sampling. However, a proof (or disproof) is beyond reach. As discussed in Section 4.4, proof by induction fails, because it does not hold that a larger covariance of the pre-activation implies a larger second raw moment of the post-activation. It remains a valuable avenue of future work to formalize the observation to a rigorous analysis.

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

## A  DETAILS OF FIGURE 1

**Graph:** The graph has 100 nodes. Define $x_j = (\frac{j}{50} - 1)\pi \in [-\pi, \pi]$, $j = 1, \ldots, 100$. The adjacency matrix is

$$A_{ij} = \begin{cases} 1, & \cos(x_i - x_j) \geq 0.9 \\ 0, & \cos(x_i - x_j) < 0.9. \end{cases}$$

As a result, each node is connected to its 14 closest neighbors. Each node also includes a self-loop, which can be interpreted as a normalization of the adjacency matrix.

**Training nodes:** The training set contains six nodes $(x_b, f_b)$ where

$$x_b = -2.5, -1.5, -0.5, 0.5, 1.5, 2.5,$$

by following the function $f(x) = \sin(\frac{\pi}{2}x + \frac{\pi}{4})$. Note that the adjacency matrix is circulant and by symmetry, a GNN will give the same prediction at $x_b = \pm\pi$. However, $f(\pi) \neq f(-\pi)$ and hence $f$ should not be considered as the function to learn. It only generates training data.

**Neural network:** The GCN has two layers with dimensions $d_0 = 100, d_1 = 100, d_2 = 1$ and scales $\sigma_w^2 = 32, \sigma_b^2 = 0$. The input feature matrix is the identity matrix $X^{(0)} = I_{100 \times 100}$. Using the identity as $X^{(0)}$ is a common practice for GNNs without node features. Here, the illustration is more interesting than using the x-coordinate as the node feature.

**Training:** The training of GCN follows standard gradient descent $\theta' = \theta - \eta\nabla_\theta\mathcal{L}$ with constant learning rate $\eta = 0.1$.

**Sampling distribution:** The example uses layer-wise sampling without replacement and uniform sampling probability $q_l(v) = 1/2$ for all nodes $v$ and layers $l$.

## B  PROOFS

### B.1  PROOF OF THEOREM 2 (GCN-NTK)

We prove by mathematical induction. Start with $l = 1$, in which case

$$f(\theta) = Z^{(1)} = \frac{\sigma_w}{\sqrt{d_0}}AX^{(0)}W^{(1)} + \sigma_b\mathbf{1}_{N \times 1}b^{(1)}$$

without using the nonlinearity $\phi$ at the end. Moreover, the output $Z^{(1)}$ has a single column. We can calculate that the NTK

$$\Theta^{(1)}(\theta) = \sum_{i=1}^{d_0}\left(\frac{\partial Z^{(1)}}{\partial W_{i1}^{(1)}}\right)\left(\frac{\partial Z^{(1)}}{\partial W_{i1}^{(1)}}\right)^T + \left(\frac{\partial Z^{(1)}}{\partial b_1^{(1)}}\right)\left(\frac{\partial Z^{(1)}}{\partial b_1^{(1)}}\right)^T$$

$$= \frac{\sigma_w^2}{d_0}\sum_{i=1}^{d_0}AX_{:i}^{(0)}(X_{:i}^{(0)})^TA^T + \sigma_b^2\mathbf{1}_{N \times 1}\mathbf{1}_{N \times 1}^T$$

$$= \sigma_w^2 AC^{(0)}A^T + \sigma_b^2\mathbf{1}_{N \times N}$$

$$= \Theta^{(1)},$$

which is independent of the parameter $\theta$. This completes the proof of the base case.

Using induction, we assume that the theorem holds up to an $l$-layer GCN, whose parameters are denoted by

$$\widetilde{\theta} = (W^{(1)}, \ldots, W^{(l)}, b^{(1)}, \ldots, b^{(l)}) \in \mathbb{R}^{\widetilde{P}}$$

for notational convenience. As we move beyond a 1-layer GCN, the covariance matrix and the NTK take limits as $d_1, \ldots, d_{l-1} \to \infty$. In particular,

$$\Theta^{(l)}(\widetilde{\theta}) = \sum_{p=1}^{\widetilde{P}}\left(\frac{\partial Z_{:1}^{(l)}}{\partial\widetilde{\theta}_p}\right)\left(\frac{\partial Z_{:1}^{(l)}}{\partial\widetilde{\theta}_p}\right)^T \to \Theta^{(l)}.$$

We now proceed to the case $l + 1$, where the GCN is

$$f(\theta) = Z^{(l+1)} = \frac{\sigma_w}{\sqrt{d_l}} A\phi(Z^{(l)})W^{(l+1)} + \sigma_b \mathbf{1}_{N \times 1} b^{(l+1)}$$

and the parameter is $\theta = (W^{(l+1)}, b^{(l+1)}, \widetilde{\theta})$. Again, note that $Z^{(l+1)}$ and $W^{(l+1)}$ have a single column and $b^{(l+1)}$ is a scalar. We write

$$\Theta^{(l+1)}(\theta) = \underbrace{\sum_{i=1}^{d_l} \left(\frac{\partial Z^{(l+1)}}{\partial W_{i1}^{(l+1)}}\right)\left(\frac{\partial Z^{(l+1)}}{\partial W_{i1}^{(l+1)}}\right)^T}_{(i)} + \underbrace{\left(\frac{\partial Z^{(l+1)}}{\partial b_1^{(l+1)}}\right)\left(\frac{\partial Z^{(l+1)}}{\partial b_1^{(l+1)}}\right)^T}_{(ii)}$$

$$+ \underbrace{\sum_{p=1}^{\widetilde{P}} \left(\frac{\partial Z^{(l+1)}}{\partial \widetilde{\theta}_p}\right)\left(\frac{\partial Z^{(l+1)}}{\partial \widetilde{\theta}_p}\right)^T}_{(iii)}.$$

Let us compute the three terms one by one. Clearly,

$$\text{Term } (i) = \frac{\sigma_w^2}{d_l} \sum_{i=1}^{d_l} A\phi(Z_{:i}^{(l)})\phi(Z_{:i}^{(l)})^T A^T$$

$$\to \sigma_w^2 A \left(\mathrm{E}_{z_i^{(l)} \sim \mathcal{N}(0, K^{(l)})}[\phi(z_i^{(l)})\phi(z_i^{(l)})^T]\right) A^T$$

$$= \sigma_w^2 A C^{(l)} A^T,$$

and

$$\text{Term } (ii) = \sigma_b^2 \mathbf{1}_{N \times 1} \mathbf{1}_{N \times 1}^T = \sigma_b^2 \mathbf{1}_{N \times N}.$$

Additionally, for each parameter $\widetilde{\theta}_p$,

$$\frac{\partial Z^{(l+1)}}{\partial \widetilde{\theta}_p} = \frac{\sigma_w}{\sqrt{d_l}} A \frac{\partial \phi(Z^{(l)})}{\partial \widetilde{\theta}_p} W^{(l+1)} = \frac{\sigma_w}{\sqrt{d_l}} A \left(\dot{\phi}(Z^{(l)}) \odot \frac{\partial Z^{(l)}}{\partial \widetilde{\theta}_p}\right) W^{(l+1)}.$$

Therefore,

$$\text{Term } (iii) = \frac{\sigma_w^2}{d_l} \sum_{p=1}^{\widetilde{P}} A \left(\dot{\phi}(Z^{(l)}) \odot \frac{\partial Z^{(l)}}{\partial \widetilde{\theta}_p}\right) W_{:1}^{(l+1)} W_{:1}^{(l+1)T} \left(\dot{\phi}(Z^{(l)}) \odot \frac{\partial Z^{(l)}}{\partial \widetilde{\theta}_p}\right)^T A^T$$

$$\to \frac{\sigma_w^2}{d_l} \sum_{p=1}^{\widetilde{P}} A \left(\dot{\phi}(Z^{(l)}) \odot \frac{\partial Z^{(l)}}{\partial \widetilde{\theta}_p}\right)\left(\dot{\phi}(Z^{(l)}) \odot \frac{\partial Z^{(l)}}{\partial \widetilde{\theta}_p}\right)^T A^T.$$

Note that all the columns of $\frac{\partial Z^{(l)}}{\partial \widetilde{\theta}_p}$ are the same; hence,

$$\left(\dot{\phi}(Z^{(l)}) \odot \frac{\partial Z^{(l)}}{\partial \widetilde{\theta}_p}\right)\left(\dot{\phi}(Z^{(l)}) \odot \frac{\partial Z^{(l)}}{\partial \widetilde{\theta}_p}\right)^T = \left(\dot{\phi}(Z^{(l)})\dot{\phi}(Z^{(l)})^T\right) \odot \left(\left(\frac{\partial Z_{:1}^{(l)}}{\partial \widetilde{\theta}_p}\right)\left(\frac{\partial Z_{:1}^{(l)}}{\partial \widetilde{\theta}_p}\right)^T\right).$$

Therefore,

$$\frac{1}{d_l} \sum_{p=1}^{\widetilde{P}} \left(\dot{\phi}(Z^{(l)}) \odot \frac{\partial Z^{(l)}}{\partial \widetilde{\theta}_p}\right)\left(\dot{\phi}(Z^{(l)}) \odot \frac{\partial Z^{(l)}}{\partial \widetilde{\theta}_p}\right)^T \to \mathrm{E}_{z_i^{(l)} \sim \mathcal{N}(0, K^{(l)})}[\dot{\phi}(z_i^{(l)})\dot{\phi}(z_i^{(l)})^T] \odot \Theta^{(l)},$$

and thus

$$\text{Term } (iii) \to \sigma_w^2 A (\dot{C}^{(l)} \odot \Theta^{(l)}) A^T.$$

Altogether, we have

$$\Theta^{(l+1)}(\theta) \to \sigma_w^2 A C^{(l)} A^T + \sigma_b^2 \mathbf{1}_{N \times N} + \sigma_w^2 A (\dot{C}^{(l)} \odot \Theta^{(l)}) A^T = \Theta^{(l+1)},$$

which completes the mathematical induction.

## B.2  PROOF OF THEOREM 3 (EVOLUTION OF GCN-NTK)

Recall that the ODE is

$$\frac{d}{dt}\begin{pmatrix} f_b(t) \\ f_c(t) \end{pmatrix} = -\frac{\eta}{N_b}\begin{pmatrix} \Theta_{bb} & \Theta_{bc} \\ \Theta_{cb} & \Theta_{cc} \end{pmatrix}\begin{pmatrix} f_b(t) - y_b \\ 0 \end{pmatrix},$$

and solving it separately for $f_b$ and $f_c$ results in

$$f_b(t) = \beta y_b + \alpha f_b(0),$$
$$f_c(t) = \Theta_{cb}\Theta_{bb}^{-1}\beta y_b + f_c(0) - \Theta_{cb}\Theta_{bb}^{-1}\beta f_b(0).$$

Note that the ground truth $y_b$ is a constant, while $f_b(0)$ and $f_c(0)$ are jointly normal with

$$\begin{pmatrix} f_b(0) \\ f_c(0) \end{pmatrix} \sim \mathcal{N}\left(0, \begin{pmatrix} K_{bb}(0) & K_{bc}(0) \\ K_{cb}(0) & K_{cc}(0) \end{pmatrix}\right).$$

Therefore, taking expectation and covariance, we obtain that the mean of $(f_b, f_c)$ has two components

$$\mu_b(t) = \beta y_b,$$
$$\mu_c(t) = \Theta_{cb}\Theta_{bb}^{-1}\beta y_b,$$

and the covariance has four components

$$K_{bb}(t) = \mathrm{E}[\alpha f_b(0)f_b(0)^T\alpha^T] = \alpha K_{bb}(0)\alpha,$$
$$K_{cb}(t) = \mathrm{E}[(f_c(0) - \Theta_{cb}\Theta_{bb}^{-1}\beta f_b(0))f_b(0)^T\alpha^T] = K_{cb}(0)\alpha - \Theta_{cb}\Theta_{bb}^{-1}\beta K_{bb}(0)\alpha,$$
$$K_{cc}(t) = \mathrm{E}[(f_c(0) - \Theta_{cb}\Theta_{bb}^{-1}\beta f_b(0))(f_c(0) - \Theta_{cb}\Theta_{bb}^{-1}\beta f_b(0))^T]$$
$$= K_{cc}(0) - \Theta_{cb}\Theta_{bb}^{-1}\beta K_{bc}(0) - K_{cb}(0)\beta\Theta_{bb}^{-1}\Theta_{bc} + \Theta_{cb}\Theta_{bb}^{-1}\beta K_{bb}(0)\beta\Theta_{bb}^{-1}\Theta_{bc},$$

with $K_{bc}(t) = K_{cb}(t)^T$.

## B.3  PROOF OF THEOREM 4 (GCN-NTK UNDER LAYER-WISE SAMPLING WITH REPLACEMENT; A.K.A., FASTGCN-NTK)

Under layer-wise sampling with replacement, the recursion of GCN becomes

$$Z^{(l)} = \sigma_b \mathbf{1}_{N \times 1} b^{(l)} + \frac{\sigma_w}{\sqrt{d_{l-1}}}\frac{1}{N_l}\sum_{j=1}^{N_l} AD_l^j X^{(l-1)}W^{(l)},$$

where $\mathrm{E}[D_l^j] = I_{N \times N}$ for all $j$ and $l$. Our objective is to compute $K^{(l)}$ as the covariance of a column of $Z^{(l)}$. Without loss of generality, we consider the $i$-th column. The GCN recursion becomes

$$z_i^{(l)} = \sigma_b \mathbf{1}_{N \times 1} b_i^{(l)} + \frac{\sigma_w}{\sqrt{d_{l-1}}}\frac{1}{N_l}\sum_{j=1}^{N_l} AD_l^j y_i^{(l)} \quad \text{where} \quad y_i^{(l)} = X^{(l-1)}W_{:,i}^{(l)}.$$

Straightforwardly,

$$K^{(l)} = \mathrm{E}\left[z_i^{(l)}z_i^{(l)^T}\right] = \sigma_b^2 \mathbf{1}_{N \times N} + \sigma_w^2 \frac{1}{N_l^2}\sum_{j=1}^{N_l}\sum_{j'=1}^{N_l} A \cdot \mathrm{E}[D_l^j C^{(l-1)} D_l^{j'}] \cdot A^T.$$

When $j \neq j'$, $D_l^j$ and $D_l^{j'}$ are independent; hence,

$$\mathrm{E}[D_l^j C^{(l-1)} D_l^{j'}] = \mathrm{E}[D_l^j]\,\mathrm{E}[C^{(l-1)}]\,\mathrm{E}[D_l^{j'}] = C^{(l-1)}.$$

When $j = j'$, $D_l^j$ takes $e_v e_v^T / p_l(v)$ with probability $p_l(v)$ for each $v \in \mathcal{V}$. Hence,

$$\mathrm{E}[D_l^j C^{(l-1)} D_l^{j'}] = \sum_{v \in \mathcal{V}} \frac{1}{p_l(v)} e_v e_v^T C^{(l-1)} e_v e_v^T = \mathrm{diag}\{\mathrm{diag}(C^{(l-1)})/p_l\}.$$

Therefore,

$$K^{(l)} = \sigma_b^2 \mathbf{1}_{N \times N} + \sigma_w^2 \frac{1}{N_l^2} A \left[ N_l(N_l - 1)C^{(l-1)} + N_l \cdot \text{diag}\{\text{diag}(C^{(l-1)})/p_l\} \right] A^T$$

$$= \sigma_b^2 \mathbf{1}_{N \times N} + \sigma_w^2 A(M_p^{(l)} \odot C^{(l-1)})A^T,$$

where

$$M_p^{(l)}(v, v') = \begin{cases} 1 + \frac{1 - p_l(v)}{p_l(v)N_l}, & v = v' \\ 1 - \frac{1}{N_l}, & v \neq v'. \end{cases}$$

The recursion for $\Theta^{(l)}$ is established analogously.

## B.4 PROOF OF THEOREM 5 (GCN-NTK UNDER LAYER-WISE SAMPLING WITHOUT REPLACEMENT)

If we use $\mathcal{V}_l$ to denote the set of sampled nodes for layer $l$, we can write

$$\sum_{v \in \mathcal{V}} A_{xv} y_i^{(l)}(v) = \mathrm{E}_{\mathcal{V}_l} \left[ \sum_{v \in \mathcal{V}_l} \frac{1}{q_l(v)} A_{xv} y_i^{(l)}(v) \right].$$

Hence, the recursion of GCN becomes

$$z_i^{(l)}(x) = \sigma_b b_i^{(l)} + \frac{\sigma_w}{\sqrt{d_{l-1}}} \sum_{v \in \mathcal{V}_l} \frac{1}{q_l(v)} A_{xv} y_i^{(l)}(v) = \sigma_b b_i^{(l)} + \frac{\sigma_w}{\sqrt{d_{l-1}}} \sum_{v \in \mathcal{V}} \frac{I_{\mathcal{V}_l}(v)}{q_l(v)} A_{xv} y_i^{(l)}(v),$$

where $I_{\mathcal{V}_l}(v)$ is the indicator function for $v \in \mathcal{V}_l$. Therefore,

$$K^{(l)}(x, x') = \mathrm{E}\left[ z_i^{(l)}(x) z_i^{(l)}(x') \right]$$

$$= \sigma_b^2 + \frac{\sigma_w^2}{d_{l-1}} \sum_{v \in \mathcal{V}} \sum_{v' \in \mathcal{V}} A_{xv} \mathrm{E}\left[ \frac{I_{\mathcal{V}_l}(v)}{q_l(v)} \frac{I_{\mathcal{V}_l}(v')}{q_l(v')} y_i^{(l)}(v) y_i^{(l)}(v') \right] A_{x'v'}.$$

Because the randomness of $\mathcal{V}_l$ comes from neighborhood sampling and the randomness of $y_i^{(l)}$ comes from random parameters, they are independent and we obtain

$$\frac{1}{d_{l-1}} \mathrm{E}\left[ \frac{I_{\mathcal{V}_l}(v)}{q_l(v)} \frac{I_{\mathcal{V}_l}(v')}{q_l(v')} y_i^{(l)}(v) y_i^{(l)}(v') \right] = \underbrace{\mathrm{E}\left[ \frac{I_{\mathcal{V}_l}(v)}{q_l(v)} \frac{I_{\mathcal{V}_l}(v')}{q_l(v')} \right]}_{M_q^{(l)}(v,v')} \underbrace{\mathrm{E}\left[ \frac{y_i^{(l)}(v) y_i^{(l)}(v')}{d_{l-1}} \right]}_{C^{(l-1)}(v,v')},$$

where $M_q^{(l)}(v, v') = 1/q_l(v)$ if $v = v'$ and $= 1$ otherwise. Hence,

$$K^{(l)}(x, x') = \sigma_b^2 + \sigma_w^2 A(x, :)(M_q^{(l)} \odot C^{(l-1)})A(x', :)^T.$$

The recursion for $\Theta^{(l)}$ is established analogously.

## B.5 PROOF OF THEOREM 6 (GCN-NTK UNDER NODE-WISE SAMPLING)

We use $\mathcal{V}_x \subset \mathcal{N}(x)$ to denote the set of sampled nodes for $x$ (in layer $l$). We drop the layer index in $\mathcal{V}_x$ to avoid notation cluttering. Under node-wise sampling, the recursion of GCN becomes

$$z_i^{(l)}(x) = \sigma_b b_i^{(l)} + \frac{\sigma_w}{\sqrt{d_{l-1}}} \sum_{v \in \mathcal{N}(x)} \frac{I_{\mathcal{V}_x}(v)}{q_x(v)} A_{xv} y_i^{(l)}(v),$$

where $I_{\mathcal{V}_x}(v)$ is the indicator function for $v \in \mathcal{V}_x$. Then,

$$K^{(l)}(x, x') = \mathrm{E}\left[ z_i^{(l)}(x) z_i^{(l)}(x') \right]$$

$$= \sigma_b^2 + \frac{\sigma_w^2}{d_{l-1}} \sum_{v \in \mathcal{N}(x)} \sum_{v' \in \mathcal{N}(x')} A_{xv} \mathrm{E}\left[ \frac{I_{\mathcal{V}_x}(v)}{q_x(v)} \frac{I_{\mathcal{V}_{x'}}(v')}{q_{x'}(v')} y_i^{(l)}(v) y_i^{(l)}(v') \right] A_{x'v'}.$$

Because the randomness of $\mathcal{V}_x$ comes from neighborhood sampling and the randomness of $y_i^{(l)}$ comes from random parameters, they are independent and we obtain

$$
\frac{1}{d_{l-1}} \operatorname{E}\left[\frac{I_{\mathcal{V}_x}(v)}{q_x(v)}\frac{I_{\mathcal{V}_{x'}}(v')}{q_{x'}(v')}y_i^{(l)}(v)y_i^{(l)}(v')\right] = \underbrace{\operatorname{E}\left[\frac{I_{\mathcal{V}_x}(v)}{q_x(v)}\frac{I_{\mathcal{V}_{x'}}(v')}{q_{x'}(v')}\right]}_{M_{xx'}^{(l)}(v,v')}\underbrace{\operatorname{E}\left[\frac{y_i^{(l)}(v)y_i^{(l)}(v')}{d_{l-1}}\right]}_{C^{(l-1)}(v,v')},
$$

where for $v \in \mathcal{N}(x)$ and $v' \in \mathcal{N}(x')$,

$$
\text{when } x = x' : \quad M_{xx'}^{(l)}(v,v') = 1/q_x(v) \text{ if } v = v' \text{ and } = 1 \text{ otherwise}
$$

$$
\text{when } x \neq x' : \quad M_{xx'}^{(l)}(v,v') = 1.
$$

Hence,

$$
K^{(l)}(x,x') = \sigma_b^2 + \sigma_w^2 A(x,:)(M_{xx'}^{(l)} \odot C^{(l-1)})A(x',:)^T.
$$

The recursion for $\Theta^{(l)}$ is established analogously.

### B.6 PROOF OF PROPOSITION 8 (PROPERTIES OF THE MASKING MATRICES)

**Property (i):** It is easy to see that $M_p^{(l)}$ can be written in the matrix form

$$
M_p^{(l)} = \mathbf{1}_{N \times N} + \frac{1}{N_l}P^{(l)} \quad \text{where} \quad P^{(l)} = \operatorname{diag}(1/p_l) - \mathbf{1}_{N \times N}.
$$

For any vector $v$,

$$
v^T P^{(l)} v = \left(\sum_{i=1}^{N} \frac{v_i^2}{p_i}\right) - \left(\sum_{i=1}^{N} v_i\right)^2.
$$

Then, by Cauchy-Schwarz,

$$
\left(\sum_{i=1}^{N} v_i\right)^2 = \left(\sum_{i=1}^{N} \sqrt{p_i}\frac{v_i}{\sqrt{p_i}}\right)^2 \leq \left(\sum_{i=1}^{N} p_i\right)\left(\sum_{i=1}^{N} \frac{v_i^2}{p_i}\right) = \left(\sum_{i=1}^{N} \frac{v_i^2}{p_i}\right),
$$

which indicates that $v^T P^{(l)} v \geq 0$. Therefore, $P^{(l)}$ is symmetric positive semi-definite (SPSD) and hence $M_p^{(l)} \succeq \mathbf{1}_{N \times N}$.

**Property (ii):** We write $M_q^{(l)}$ in the matrix form

$$
M_q^{(l)} = \mathbf{1}_{N \times N} + Q^{(l)} \quad \text{where} \quad Q^{(l)} = \operatorname{diag}(1/q_l - 1).
$$

Because $q_l(v) \in (0,1]$ for all $v$, $Q^{(l)}$ is SPSD and hence $M_q^{(l)} \succeq \mathbf{1}_{N \times N}$.

**Property (iii):** Because $M_p^{(l)} - \mathbf{1}_{N \times N} \succeq \mathbf{0}_{N \times N}$ and $M_1^{(l)} - \mathbf{1}_{N \times N} \succeq \mathbf{0}_{N \times N}$, the property follows from the Schur Product Theorem (the Hadamard product of two SPSD matrices is SPSD).

**Property (iv):** When $q_l = N_l p_l$, we have

$$
M_p^{(l)} - M_q^{(l)} = \frac{1}{N_l}(P^{(l)} - N_l Q^{(l)}) = N_l \cdot I_{N \times N} - \mathbf{1}_{N \times N} =: \Delta.
$$

Because $\Delta$ is symmetric, it has $N$ real eigenvalues. For the vector of all ones, we have $\Delta\mathbf{1}_{N \times 1} = (N_l - N)\mathbf{1}_{N \times 1}$. For any vector $v \perp \mathbf{1}_{N \times 1}$, we have $\Delta v = N_l v$. Hence, $\Delta$ has one eigenvalue equal to $N_l - N$ and $N - 1$ eigenvalues equal to $N_l$.

### B.7 PROOF OF THEOREM 9 (GRAPHSAGE-NTK)

We rewrite the GraphSAGE recursion by replacing $X^{(l-1)}$ with $\phi(Z^{(l-1)})$:

$$
Z^{(l)} = \underbrace{\frac{\sigma_w}{\sqrt{d_{l-1}}}\phi(Z^{(l-1)})W_1^{(l)}}_{(i)} + \underbrace{\frac{\sigma_w}{\sqrt{d_{l-1}}}A\phi(Z^{(l-1)})W_2^{(l)}}_{(ii)} + \underbrace{\sigma_b\mathbf{1}_{N \times 1}b^{(l)}}_{(iii)}.
$$

The covariances of the first column of the three terms are: (i): $\sigma_w^2 C^{(l-1)}$; (ii): $\sigma_w^2 A C^{(l-1)} A^T$; (iii): $\sigma_b^2\mathbf{1}_{N \times N}$. Hence,

$$
K^{(l)} = \sigma_w^2 C^{(l-1)} + \sigma_w^2 A C^{(l-1)} A^T + \sigma_b^2\mathbf{1}_{N \times N}.
$$

The proof for $\Theta^{(l)}$ is analogous.

### B.8 Proof of Theorem 10 (GraphSAGE-NTK under node-wise sampling)

We rewrite the GraphSAGE recursion by replacing $X^{(l-1)}W^{(l)}$ with $Y^{(l)}$ and toggle only the node $x$ and the $i$-th column:

$$
z_i^{(l)}(x) = \underbrace{\frac{\sigma_w}{\sqrt{d_{l-1}}} y_i^{(l)}(x)}_{(i)} + \underbrace{\frac{\sigma_w}{\sqrt{d_{l-1}}} \sum_{v \in \mathcal{N}(x)} \frac{I_{\mathcal{V}_x}(v)}{q_x(v)} A_{xv} y_i^{(l)}(v)}_{(ii)} + \underbrace{\sigma_b b_i^{(l)}}_{(iii)}.
$$

The covariance $\mathrm{E}[z_i^{(l)}(x) z_i^{(l)}(x')]$ has three terms: (i): $\sigma_w^2 C^{(l-1)}(x, x')$; (ii): $\sigma_w^2 A(x,:)(M_{xx'}^{(l)} \odot C^{(l-1)}) A(x',:)^T$; (iii): $\sigma_b^2$. Hence,

$$
K^{(l)}(x, x') = \sigma_w^2 C^{(l-1)}(x, x') + \sigma_w^2 A(x,:)(M_{xx'}^{(l)} \odot C^{(l-1)}) A(x',:)^T + \sigma_b^2.
$$

The proof for $\Theta^{(l)}(x, x')$ is analogous.

## C Formulas for $C^{(l)}$ and $\dot{C}^{(l)}$

Recall that

$$
C^{(l)} = \mathrm{E}_{z_i^{(l)} \sim \mathcal{N}(0, K^{(l)})} [\phi(z_i^{(l)}) \phi(z_i^{(l)})^T],
$$

$$
\dot{C}^{(l)} = \mathrm{E}_{z_i^{(l)} \sim \mathcal{N}(0, K^{(l)})} [\dot{\phi}(z_i^{(l)}) \dot{\phi}(z_i^{(l)})^T].
$$

When $\phi$ is ReLU, $C^{(l)}$ and $\dot{C}^{(l)}$ are (half of) the arc-cosine kernel $k_n$ of order $n = 1$ and $n = 0$, respectively, as defined in Cho & Saul (2009). The arc-cosine kernels have closed-forms for integer orders, albeit being increasingly complex as $n$ increases. Specifically, using the notations in Cho & Saul (2009),

$$
\mathrm{E}_{z \sim \mathcal{N}(0, K)} [\phi(z(x)) \phi(z(x'))^T] = \frac{1}{2} k_1(K^{\frac{1}{2}} e_x, K^{\frac{1}{2}} e_{x'}) = \frac{1}{2\pi} \|K^{\frac{1}{2}} e_x\| \|K^{\frac{1}{2}} e_{x'}\| J_1(\theta)
$$

and similarly for $\mathrm{E}_{z \sim \mathcal{N}(0, K)} [\dot{\phi}(z(x)) \dot{\phi}(z(x'))^T]$, where $J_n$ has closed forms. Substituting the closed forms and simplifying, we obtain

$$
C_{xx'}^{(l)} = \frac{1}{2\pi} \sqrt{K_{xx}^{(l)} K_{x'x'}^{(l)}} \left( \sin \theta_{xx'}^{(l)} + (\pi - \theta_{xx'}^{(l)}) \cos \theta_{xx'}^{(l)} \right), \tag{27}
$$

$$
\dot{C}_{xx'}^{(l)} = \frac{1}{2\pi} \left( \pi - \theta_{xx'}^{(l)} \right), \tag{28}
$$

where

$$
\theta_{xx'}^{(l)} = \arccos \left( \frac{K_{xx'}^{(l)}}{\sqrt{K_{xx}^{(l)} K_{x'x'}^{(l)}}} \right). \tag{29}
$$

When $\phi$ is erf, Williams (1996) computed $C^{(l)}$:

$$
C_{xx'}^{(l)} = \frac{2}{\pi} \arcsin \left( \frac{2 K_{xx'}^{(l)}}{\sqrt{(1 + 2 K_{xx}^{(l)})(1 + 2 K_{x'x'}^{(l)})}} \right), \tag{30}
$$

$$
\dot{C}_{xx'}^{(l)} = \frac{4}{\pi} \left( (1 + 2 K_{xx}^{(l)})(1 + 2 K_{x'x'}^{(l)}) - (2 K_{xx'}^{(l)})^2 \right)^{-\frac{1}{2}}, \tag{31}
$$

while $\dot{C}^{(l)}$ is straightforward by following the expectation integral and noting that $\dot{\phi}$ is squared exponential.

# D    Conjecture of $C$ and $\dot{C}$

Let

$$C_i = \mathrm{E}_{z_i \sim \mathcal{N}(0, K_i)}[\phi(z_i)\phi(z_i)^T] \quad \text{and} \quad \dot{C}_i = \mathrm{E}_{z_i \sim \mathcal{N}(0, K_i)}[\dot{\phi}(z_i)\dot{\phi}(z_i)^T] \quad \text{for } i = 1, 2.$$

It is suspected that if $K_1 \succeq K_2$, then $C_1 \succeq C_2$ and $\dot{C}_1 \succeq \dot{C}_2$. In the following, we give an example to show that neither conclusion holds.

Let

$$K_1 = \begin{bmatrix} 2.58 & 0.83 \\ 0.83 & 0.62 \end{bmatrix} \quad \text{and} \quad K_2 = \begin{bmatrix} 1.52 & 0.76 \\ 0.76 & 0.61 \end{bmatrix}.$$

One can verify that $K_1 \succeq K_2$. However,

$$\text{when } \phi = \mathrm{ReLU}, \quad \text{eigenvalues of } C_1 - C_2 = -0.00172870, \ 0.53672870$$
$$\text{eigenvalues of } \dot{C}_1 - \dot{C}_2 = -0.03084086, \ 0.03084086$$
$$\text{when } \phi = \mathrm{erf}, \quad \text{eigenvalues of } C_1 - C_2 = -0.01530613, \ 0.10825164$$
$$\text{eigenvalues of } \dot{C}_1 - \dot{C}_2 = -0.17231565, \ 0.06827739.$$

