# OpenReview forum: "Neighborhood Sampling Does Not Learn the Same Graph Neural Network"
_ICLR.cc/2026/Conference — Submitted to ICLR 2026_

### Official Review · Reviewer_AFpS · 2025-10-28

**Soundness:** 2
**Presentation:** 2
**Contribution:** 2
**Rating:** 2
**Confidence:** 3

**Summary:**

This paper provides a theoretical analysis of neighborhood sampling in GNNs through the lens of neural tangent kernels (NTKs), which connect the training dynamics of infinitely wide networks to Gaussian processes (GPs). By examining several established sampling strategies and their corresponding posterior GPs, the paper shows that while all posteriors converge in the infinite-sample limit, they differ under limited sampling. This paper further demonstrates that the posterior covariances are incomparable, consistent with empirical findings that no single sampling method universally outperforms others.

**Strengths:**

1. This paper derives the posterior inference for evolving GNN-GPs, providing a novel and rigorous extension of prior GNN-GP theory that had only been sporadically explored.

2. This paper offers a clear comparative analysis of major neighborhood sampling methods, revealing their convergence behaviors and theoretical incomparability, which aligns with empirical GNN performance differences.

3. This paper introduces a general, programmable framework for constructing GNTKs in arbitrary GNNs, extending composability theory to graph domains and demonstrating its utility through GraphSAGE examples.

**Weaknesses:**

**1. Theoretical motivation and methodological justification**

This paper conducts a theoretical study on neighborhood sampling.

1). However, it remains unclear whether or how prior works provide theoretical explanations of neighborhood sampling. Does this paper fill any specific existing gaps? The authors should clarify the related works and what the precise theoretical gaps are.

2). Why NTK is chosen as the main analytical tool? Have alternative analyses (e.g., gradient norm bounds, variance reduction perspectives) been considered? Moreover, the infinite-width assumption in NTK may average out per-neuron gradient noise, potentially obscuring the distinct effects of different sampling methods. Could this limit the conclusions in this paper when comparing different neighbor sampling methods?

**2. Practical meaning and theoretical contribution**

1). The paper’s extension to posterior inference needs further clarification of its practical meaning. What specific GNN phenomena does this posterior analysis help explain?

2). The extension of GNTK to multiple neighborhood sampling schemes is interesting, but what are the key challenges in making this extension? Does the analysis introduce any conceptual or theoretical breakthroughs beyond the existing GNTK framework?

3). Finally, the conclusion that different sampling methods are “uncomparable” does not seem to provide new theoretical insights or trade-offs. Please elaborate on the novelty of this finding.

**3. Minor suggestion (Figure 1)**

For Figure 1, please clarify the meaning of the vertical axis (which is not described in Appendix A or the main text). The title states “neighborhood sampling drives the GCN-GP to evolve faster to the limit,” but it is unclear how to visually interpret “faster.” If t denotes time, the differences among 𝑡=0,10,100 appear small, as the blue and red curves nearly coincide at t=100. Can this “faster” convergence be quantified or made more explicit in the figure or text?

**4. Interpretation of the main conclusion**

Given the conclusion that “neighborhood sampling does not learn the same graph neural network,” what are the practical implications of learning different GNNs? Does this lead to measurable changes in test accuracy or any other metrics? Please clarify what specific differences these “different learned GNNs” induce. Could these distinctions be demonstrated empirically on real datasets?

**Questions:**

See weaknesses.

If the authors can address these questions, I am willing to raise my score.

---

### Official Review · Reviewer_EbBb · 2025-10-28

**Soundness:** 2
**Presentation:** 2
**Contribution:** 1
**Rating:** 2
**Confidence:** 3

**Summary:**

The paper studies neighborhood sampling in GNNs through the lens of Gaussian processes and neural tangent kernels. It derives GNTKs and posterior means and covariances of the associated Gaussian processes for several GNN sampling strategies, aiming to understand how sampling affects learning dynamics and predictive performance.

**Strengths:**

- The paper studies neighborhood sampling in GNNs through the lens of Gaussian processes and neural tangent kernels (NTKs). It derives GNTKs and posterior means covariances of the associated Gaussian processes (GPs) for several GNN sampling strategies, aiming to understand how sampling affects learning dynamics and predictive performance. This is a relevant topic.
- The analysis of the posterior covariance is quite relevant since, as the authors note, the posterior covariance is a lower bound of the mean squared prediction error for GPs. However, this is not emphasized enough in the paper.
- The conclusion that the posterior mean is unbiased in the infinite-width limit for FastGCN, even though FastGCN is biased, is intriguing and perhaps worthy of further exploration.

**Weaknesses:**

The topic is relevant, but the analysis is mostly mechanical. The paper presents a sequence of derivations for different setups without developing a clear theoretical message or providing intuition about what these results reveal. There is no unifying perspective on the role of sampling, and when the discussion turns to finite-width or finite-sample regimes, the authors simply note that these are difficult to analyze. The results appear to be technically correct, but are a quite shallow exploration of the problem.

Major comments:

- The derivations do not lead to conceptual insights. The main technical conclusion is that as the number of samples tend to infinity, the GNTK converges to the GNTK of the same architecture without sampling. This is unsurprising as the considered sampling-based GNNs themselves converge to the corresponding GNNs without sampling, as noted in the original references. The paper would benefit from a clearer discussion of what the theoretical results mean for learning or generalization.

- The structure reads as a disconnected, somewhat random assortment of GNN sampling algorithms rather than the analysis of a more general unified sampling framework.

- The paper stops short of addressing or interpreting both the finite-width setting but more importantly the finite-sample setting, which limits its practical relevance.

- The use of the posterior covariance as a lower bound on predictive performance should be better emphasized. Explicitly stating this lower bound for the different sampling strategies and carrying out a more careful analysis of these lower bounds’ implications would greatly improve the paper.

**Questions:**

N/A

---

### Official Review · Reviewer_KHc1 · 2025-10-31

**Soundness:** 3
**Presentation:** 3
**Contribution:** 2
**Rating:** 4
**Confidence:** 4

**Summary:**

The paper studies the convergence of neighborhood sampling on GNNs. The paper introduces a study in the neural tangent kernel -- which is an analogous analysis which looks at the infinitely wide neural network associated with it. The problem is therefore recasted as a Gaussian processes, and different standard GNN architectures are considered.

**Strengths:**

The paper is well written, and it covers a relevant topic which is neighborhood sampling and efficient training of GNNs,

**Weaknesses:**

The main issues with the paper are it's novelty and its relevance.

Regarding novelty, the same analysis has already been covered in
"For graphs, the NTK becomes a GNTK (Du et al., 2019;
Huang et al., 2022; Krishnagopal & Ruiz, 2023) and it governs the evolution of a GNNGP (Niu
et al., 2023), which is the infinite-width counterpart of a GNN."
What is the main advantage of this work?
I understand that the authors do an individual characterization of each GNN type in Table 1, but this is esoteric and with little use in practice.

Regarding the relevance of the work, this paper introduces little to none practical implications, and it is therefore very difficult to asses its relevance. What are the real world implications of the work? Not learning the same GNN remains of little use given that "different GNNs" might (and do in practice) evaluate to the same test error. And, they are therefore equally good.

**Questions:**

Can the authors add the missing citations:

Graph neural networks (GNNs) are widely used models (Zhou et al., 2020; Wu et al., 2021) for
graph-structured data, such as financial transaction networks, power grids, and molecules and crystals. They encode the relational information present in the data through message passing (Gilmer
et al., 2017) on the graph and support a wide array of tasks, including predicting node and graph
properties, generating novel graphs, and forecasting interrelated time series.

Also, the authors are missing citations to the whole Graphon -  Transferability line of work of authors like Luana Ruiz, Ron Levie, Sohir Maskey and Soledad Villar to name a few. I strongly suggest including a parragraph with their relevant works.

**Details Of Ethics Concerns:**

No.

---

### Official Review · Reviewer_8FHy · 2025-11-01

**Soundness:** 3
**Presentation:** 3
**Contribution:** 3
**Rating:** 4
**Confidence:** 3

**Summary:**

The paper does an interesting and important theoretical study of the effect of different graph sampling techniques on the convergence of graph algorithms. Building on the foundations of Neural networks as Gaussian Process (GP) and Neural Tangent Kernel (NTK), the authors analyze how the NTK and the covariance function evolve over time, which gives us an idea on their convergence. The main highlight is Theorem 7 and the following discussion, which concludes two major things: 1) At the infinite sampling limit, all sampling methods converge to the same posterior distribution. 2) There is a theoretical limitation from this perspective that two sampling strategies cannot be compared.

**Strengths:**

- The paper analyses an important aspect in GNNs, that how sampling affects the learned network function and the convergence of the algorithm.
- Such insight is valuable to the GCN community.

**Weaknesses:**

Considering that Theorem 7 and the subsequent discussion constitute the core contribution of this paper, while it is a knowledge in itself that sampling methods are theoretically incomparable in general. However, this statement alone does not yield practical guidance for choosing a sampling algorithm in practice. In that regard, I would be particularly interested in further analyses along the following directions:

**While sampling algorithms might be incomparable in general, can they be meaningfully compared under certain structural assumptions on the graph (i.e. their adjacency matrices)?**

- e.g., does a specific sampling algorithm perform better for graphs with high vs. low clustering coefficient?

- do some sampling strategies work better in heterophilic vs. homophilic graphs?

I encourage the authors to explore these scenarios more deeply. There may not exist a universally optimal sampling method for all graphs, but certain methods may be more suitable for particular graph classes. With answers to these questions, I believe the analysis in the paper would be more complete.

At this point, my evaluation is borderline. I will reconsider my score based on the authors’ response and the feedback from the other reviewers.

**Questions:**

-

---

### Meta-Review · Area_Chair_CtHp · 2026-01-07

**Summary:**

The authors study the training dynamics of graph neural networks in the NTK regime, under neighborhood sampling. The authors reached several interesting results, including poster converging to the no-sampling counterpart from the GP point of view, as well behaviour under biased samplers.

Overall, the reviewers were mostly negative towards this paper, finding several weaknesses including:
- Questionable novelty, the results being incremental
- Results lacking intuition
- Unclear practical implications
- Unsurprising conclusions
- Presentation clarity

While I tend to sympathize with the reviewers on this one, I'm still disappointed the authors did not provide a rebuttal to continue the discussion, as I could have learned more about what the authors was actually hoping to achieve with this work. I hope the authors would still consider addressing the criticisms here before resubmitting in the future.

As for this ICLR, I would have to recommend reject.

**Reviewer Concerns:**

- Questionable novelty, the results being incremental
- Results lacking intuition
- Unclear practical implications
- Unsurprising conclusions
- Presentation clarity

The authors did not provide a rebuttal, and therefore none of the concerns were addressed.

**Reviewer Scores:**

8FHy: 4
KHc1: 4
EbBb: 2
AFpS: 2

Given the authors did not provide a rebuttal, I believe the scores will remain the same.

---

### Decision · Program_Chairs · 2026-01-26

Reject